# USP2 Mitigates Reactive Oxygen Species-Induced Mitochondrial Damage via UCP2 Expression in Myoblasts

**DOI:** 10.3390/ijms252211936

**Published:** 2024-11-06

**Authors:** Hiroshi Kitamura, Masaki Fujimoto, Mayuko Hashimoto, Hironobu Yasui, Osamu Inanami

**Affiliations:** 1Laboratory of Disease Models, School of Veterinary Medicine, Rakuno Gakuen University, Ebestsu 069-8501, Japan; m-fujimoto@rakuno.ac.jp; 2Laboratory of Immunology, Faculty of Pharmacy, Osaka Ohtani University, Osaka 584-8540, Japan; hasimomayu@osaka-ohtani.ac.jp; 3Laboratory of Radiation Biology, Graduate School of Veterinary Medicine, Hokkaido University, Sapporo 060-0818, Japan; yassan@vetmed.hokudai.ac.jp (H.Y.); inanami@vetmed.hokudai.ac.jp (O.I.)

**Keywords:** ubiquitin-specific protease 2, myoblasts, mitochondria, ROS, uncoupling protein 2, PGC1α

## Abstract

Ubiquitin-specific protease 2 (USP2) maintains mitochondrial integrity in culture myoblasts. In this study, we investigated the molecular mechanisms underlying the protective role of USP2 in mitochondria. The knockout (KO) of the *Usp2* gene or the chemical inhibition of USP2 induced a robust accumulation of mitochondrial reactive oxygen species (ROS), accompanied by defects in mitochondrial membrane potential, in C2C12 myoblasts. ROS removal by N-acetyl-L-cysteine restored the mitochondrial dysfunction induced by USP2 deficiency. Comprehensive RT-qPCR screening and following protein analysis indicated that both the genetic and chemical inhibition of USP2 elicited a decrease in uncoupling protein 2 (UCP2) at mRNA and protein levels. Accordingly, the introduction of a *Ucp2*-expressing construct effectively recovered the mitochondrial membrane potential, entailing an increment in the intracellular ATP level in *Usp2*KO C2C12 cells. In contrast, USP2 deficiency also decreased peroxisome proliferator-activated receptor γ coactivator 1α (PGC1α) protein in C2C12 cells, while it upregulated *Ppargc1a* mRNA. Overexpression studies indicated that USP2 potentially stabilizes PGC1α in an isopeptidase-dependent manner. Given that PGC1α is an inducer of UCP2 in C2C12 cells, USP2 might ameliorate mitochondrial ROS by maintaining the PGC1α–UCP2 axis in myoblasts.

## 1. Introduction

Loss of muscle mass has a critical influence on quality of life. For instance, age-dependent loss of muscle mass, sarcopenia, is an atypical geriatric disease with adverse outcomes: frailty, dysmobility, and mortality [1]. The maintenance of muscle progenitor cells, including satellite cells and myoblasts, impedes the progression of sarcopenia [2]. Oxidative stress is believed to damage muscle progenitor cells. For instance, hydrogen peroxide causes apoptosis and the cytoplasmic distribution of p21 in myoblasts rather than in myotubes [3]. Moreover, excessive reactive oxygen species (ROS) severely inhibit muscle regeneration by inhibiting myocyte differentiation from progenitor cells [4,5]. Notably, satellite cells from elderly people exhibited higher intracellular ROS compared with younger people, resulting in impaired mitochondrial activity [6]. Therefore, managing intracellular ROS levels of muscle progenitor cells might be an effective therapeutic intervention for muscular atrophy.

Mitochondria are among the primary sources of ROS under both physiological and pathological conditions [7]. Under physiological conditions, mitochondria generate moderate levels of ROS, mainly from respiratory chain complexes I and III [8]. Pathological conditions, such as hyperglycemia and cardiomyopathy, perturbate the mitochondrial respiratory chain and culminate in excessive ROS production [9,10]. Superoxide is produced in the mitochondria and is converted into hydrogen peroxide by superoxide dismutase (SOD) [11]. Hydrogen peroxide is eventually quenched by catalase, the glutathione peroxidase (Gpx)–glutathione reductase (GR) system, or the peroxiredoxin (Prdx)–thioredoxin (Trx)–thioredoxin reductase (TrxR) system [12]. Uncoupling proteins (UCPs) in the inner mitochondrial membrane are also thought to minimize mitochondrial ROS generation [13]. UCPs are carrier proteins that induce proton leak and the attenuation of the mitochondrial membrane potential, resulting in the suppression of ROS production [14]. Of the three canonical UCPs, UCP2 protects against ROS in various cells, including pancreatic β-cells, cardiomyocytes, and macrophages [14,15,16]. Although UCP2 is ubiquitously expressed in multiple tissues, including skeletal muscle and adipose tissue, under normal conditions [17], the expression of the *Ucp2* gene is also under the control of transcription factors and co-activators, including peroxisomal activator-activated receptors (PPARs) and PPARγ coactivator 1α (PGC1α) [18].

Emerging evidence indicates that the excess accumulation of mitochondrial ROS severely encumbers mitochondrial function. Pathological content of mitochondrial ROS induces changes in mitochondrial morphology, oxidative phosphorylation, and mitochondrial transcription factor A levels in renal proximal tubule epithelial cells [19]. Moreover, hypoxia-induced mitochondrial ROS interrupt mitochondrial gene expression and ATP synthesis in pig sperm [20]. Mitochondrial ROS also provokes mitochondrial DNA mutation, which is associated with tissue aging [21]. Although adequate levels of ROS function as the redox signal to maintain the viability of stem cells, the prevention of excessive accumulation of ROS is necessary to sustain an efficient number of stem cells for regeneration [22]. To support this idea, Minet and Gaster reported that senescent culture human satellite cells displayed more prominent ROS accompanying an increase in mitochondrial oxidative phosphorylation [23]. The author hypothesized that accumulated ROS consequently cause mitochondrial damage at a later time. Therefore, preventing mitochondrial damage caused by ROS might be a potential therapeutic approach for regenerative medicine and geriatrics.

Ubiquitination and deubiquitination, which are counteractive protein modifications, are catalyzed by ubiquitin ligase and deubiquitinating enzymes (DUBs), respectively [24,25]. Ubiquitin-specific proteases (USPs) are the largest subfamily of DUBs, which consist of 58 members in vertebrates [26]. USP2 is a widely expressed USP with several molecular targets, including receptors, intracellular signal adaptors, and transcriptional regulators [27]. To date, we and others have demonstrated that USP2 controls energy homeostasis at local and systemic levels [28,29,30,31]. For example, USP2 in ventromedial hypothalamic neurons attenuates aberrant increases in blood glucose by mitigating glycogenolysis in the liver, whereas hepatic USP2 regulates gluconeogenesis and diurnal glucose metabolism in the liver [28,31]. Additionally, USP2 directly or indirectly maintains the mitochondrial ATP synthesis of myoblasts, sperm, and neural cells [31,32,33]. With regards to myoblasts, the chemical and genetic ablation of USP2 rapidly induces ROS accumulation, with accompanying defects in mitochondrial ATP synthesis [33]. Therefore, USP2 is likely to sustain mitochondrial respiration in myoblasts via the removal of ROS. In this study, we aimed to verify the involvement of ROS in the USP2-influenced mitochondrial dysfunction of myoblasts. To this end, we used C2C12 myoblast cells derived from regenerative muscle of C3H mice [34], which are widely used in myoblast research [33,35,36,37]. We also investigated the molecular mechanisms underlying ROS accumulation in USP2-deficient myoblasts.

## 2. Results

### 2.1. Mitochondrial ROS Are Involved in Mitochondrial Dysfunction in Usp2KO C2C12 Cells

We previously demonstrated that the knockout of the *Usp2* gene causes mitochondrial dysfunction in C2C12 myoblasts [33]. Since mitochondrial ROS markedly accumulated in *Usp2*KO C2C12 cells, we speculated that ROS accumulation might cause impaired ATP synthesis in *Usp2*KO C2C12 cells [33]. To verify this possibility, we first alleviated ROS accumulation in *Usp2*KO C2C12 cells by treating them with N-acetyl-L-cysteine (NAC) for 8 h. Figure 1A,B show mitochondrial ROS, which are visualized with MitoSOX Red in *Usp2*KO and control C2C12 cells. As previously reported [33], *Usp2*KO C2C12 cells displayed an intense MitoSOX signal compared with the vehicle-treated control C2C12 cells. NAC treatment reduced ~27% of the fluorescent signals in Usp2KO cells. Thus, NAC treatment effectively reduced ROS accumulation in *Usp2*KO C2C12 cells.

We next assessed whether the NAC treatment weakened *Usp2* deficiency-elicited mitochondrial dysfunction. The mitochondrial membrane potential, verified by tetramethylrhodamine methyl ester (TMRM) staining, was significantly lower in *Usp2*KO C2C12 cells than in control C2C12 cells (Figure 1C). NAC treatment partially recovered the mitochondrial membrane potential in *Usp2*KO C2C12 cells. Correspondingly, NAC treatment remarkably recovered the decrease in intracellular ATP in *Usp2*KO C2C12 cells (Figure 1D). Therefore, ROS damage the mitochondria of *Usp2*KO C2C12 cells, resulting in impaired ATP production.

### 2.2. Mitochondrial ROS Are Involved in Mitochondrial Dysfunction in ML364-Treated C2C12 Cells

ML364 is the most popular USP2 chemical inhibitor and has been used for USP2 blockade in vivo and in vitro [31,38]. Similar to *Usp2* gene knockout, ML364 treatment caused mitochondrial dysfunction accompanied by robust ROS accumulation in C2C12 cells [33]. Thus, we also investigated the effects of NAC on ML364-elicited mitochondrial dysfunction. Figure 2A,B show that treatment with 10 μM ML364 for 8 h substantially increased mitochondrial ROS. Since this ML364 treatment did not increase lactate dehydrogenase (LDH) content in the culture supernatant of C2C12 cells, this condition is unlikely to be toxic to C2C12 cells (Appendix A). Although NAC did not completely reverse ROS accumulation in ML364-treated cells, NAC significantly abated mitochondrial ROS levels (Figure 2A,B). Correspondingly, NAC significantly attenuated the suppressive effects of ML364 on the mitochondrial membrane potential (Figure 2C). Moreover, NAC partially restored the ML364-induced decrease in intracellular ATP (Figure 2D). Hence, ROS contribute to ML364-elicited mitochondrial damage in C2C12 cells.

### 2.3. USP2 Maintains UCP2 Expression at mRNA and Protein Levels in C2C12 Cells

Since mitochondrial respiratory complexes do not alter activity in *Usp2*KO C2C12 cells [33], the impairment of the respiratory chain is unlikely to be a major cause of excess accumulation of mitochondrial ROS in *Usp2*KO C2C12 cells. Thus, we assumed that USP2 deficiency causes expressional defects of certain antioxidative molecules. To explore this, we performed a comprehensive reverse transcription quantitative polymerase chain reaction (RT-qPCR) screening of 22 genes which can be postulated to repress ROS accumulation (Figure 3A). The expression levels of two genes was significantly (*Prdx2* and *Ucp2*; *p* < 0.01) lower in *Usp2*KO C2C12 cells than in control C2C12 cells, whereas that of three genes was higher in *Usp2*KO cells (*Gsta4*, *Hmox1*, and *Sod1*; *p* < 0.01) (Figure 3A, left). ML364 treatment influenced the expression of a larger number of genes: four genes (*Gsta4*, *Hmox1*, *Nfe2l2*, and *Prdx6*) were significantly up-regulated, and four genes (*Glrx*, *Prdx1*, *Selenos*, and *Ucp2*) were significantly down-regulated (*p* < 0.01) (Figure 3A, right). In this screening, *Ucp2* was the only gene whose expression level was more than 2-fold downregulated by both the genetic and chemical inhibition of USP2 (Figure 3B). Further RT-qPCR analysis using more replicates confirmed a decrement greater than 50% in *Ucp2* mRNA due to Usp2 deficiency in C2C12 cells (Figure 3C). Accordingly, ML364 also repressed *Ucp2* mRNA to a similar degree (Figure 3D). We also determined the UCP2 protein levels in *Usp2*KO cells. As with the mRNA levels, *Usp2*KO C2C12 cells displayed ~40% lower UCP2 protein levels than the control C2C12 cells (Figure 3E). Similarly, ML364 strongly decreased the UCP2 protein levels in C2C12 cells (Figure 3F).

We also checked the activities of canonical antioxidative enzymes in *Usp2*KO C2C12 cells and found that the activities of SOD, Gpx, GR, and TrxR were not distinguishable between control and *Usp2*KO C2C12 cells (Figure 3G). Similarly, none of them was modulated by ML364 (10 μM, 8 h) treatment (Figure 3H). Therefore, USP2 did not promote the overall activities of SOD, Gpx, GR, and TrxR in C2C12 myoblasts.

### 2.4. Decrement in UCP2 Seems to Contribute to USP2 Deficiency-Elicited Mitochondrial Dysfunction

To evaluate whether UCP2 contributes to USP2-dependent mitochondria integrity, we introduced a lentivirus encoding the *Ucp2* gene into *Usp2*KO or control C2C12 cells. Abundant *Ucp2* mRNA was observed in *Ucp2*-expressing Usp2KO and control C2C12 cells (Figure 4A). Moreover, the UCP2 protein levels between *Ucp2*-expressing *Usp2*KO and C2C12 cells were comparable (Figure 4B). Although the introduction of the *Ucp2*-expressing construct failed to reverse the excess accumulation of mitochondrial ROS in *Usp2*KO C2C12 cells, the overexpression of *Ucp2* significantly decreased ROS in these cells (~33%; Figure 4C). The *Ucp2*-expressing *Usp2*KO C2C12 cells exhibited an apparent increase in the mitochondrial membrane potential (Figure 4D). Additionally, the introduction of *Ucp2* reversed the decrease in intracellular ATP in *Usp2*KO C2C12 cells (Figure 4E). Collectively, USP2 maintains mitochondria integrity via UCP2 expression.

### 2.5. USP2 Stabilizes PGC1α in an Isopeptidase-Dependent Manner

PGC1α is known to potentiate *Ucp2* mRNA in C2C12 myotubes [39]. Thus, we investigated whether USP2 controls PGC1α protein levels. The abundance of *Ppargc1a* mRNA, which encodes PGC1α, was more than 13-fold higher in *Usp2*KO cells than in control C2C12 cells (Figure 5A). In sharp contrast, nuclear PGC1 protein levels were relatively low in *Usp2*KO C2C12 cells (Figure 5B). Accordingly, an overexpression study using 293FT cells demonstrated that the introduction of USP2 increased the accumulation of Halo-tagged PGC1α in the nucleus (Figure 5C). These results suggest that USP2 increases the PGC1α protein levels via a post-transcriptional process.

Since USP2 stabilizes several proteins by deubiquitination [40], we tested whether USP2 delayed the degradation of the PGC1α protein. Similar to the results of Figure 5C, the PGC1α protein content was remarkably lower in mock-transfected 293FT cells than in *Usp2*-transfected cells before cycloheximide application (Figure 5D). After cycloheximide addition, the PGC1α protein in mock-transfected 293FT cells was rapidly degraded, decreasing to less than 5% at 5 h (Figure 5D). However, the degradation of PGC1α was slower in *Usp2*-transfected cells: more than 70% of PGC1α was retained at 5 h, and ~37% was still visible even at 10 h.

Next, we evaluated the ubiquitin protease activity of USP2 in the control of PGC1α stability. Since the substitution of the 276th amino acid residue of USP2 from cysteine to alanine suppresses ubiquitin isopeptidase activity, we assessed the effects of the C276A mutant on PGC1α levels. In contrast to a wild-type USP2 protein, the C276A mutant failed to maintain PGC1α in 293FT cells in the presence of cycloheximide (Figure 5E). Given that the protein levels of USP2 or C276A were comparable in *USP2*- and C276A-transfected cells, USP2 modulates PGC1α levels via ubiquitin isopeptidase activity.

Finally, we examined whether USP2 directly digested the lysine48 (K48)-linked polyubiquitin chains on PGC1α, since K48-linked polyubiquitin chanins represent a canonical signal for proteasome degradation [41]. As shown in Figure 5F, an immunoprecipitation–Western blot analysis revealed that the overexpression of GFP-tagged USP2 strongly digested the K48-linked polyubiquitin chains of Halo-tagged PGC1α (see lanes for immunoprecipitants). However, the amount of PGC1α in the immunoprecipitant was smaller in the *GFP*-expressing control cells than in the *USP2-GFP*-transfected cells. Therefore, USP2 seems to directly digest the polyubiquitin chain on PGC1α.

## 3. Discussion

It was previously shown that L6 myoblasts undergo differentiation into myotubes due to expressional changes in USP2, indicating the modulatory role of USP2 in myogenesis [42]. We recently observed that the genetic or chemical inhibition of USP2 provoked the accumulation of mitochondrial ROS in C2C12 myoblasts [33]. This finding suggests that USP2 also maintains cellular function in myoblasts. In agreement with this, our preliminary experiments showed that the inhibition of USP2 also potentiated mitochondrial ROS production in myotubes. Thus, USP2 seems to prevent ROS production in mitochondria in the muscle cell lineage. Given that the chemical blockade of USP2 brought about ROS accumulation in hypothalamic neurons, leading to the activation of the efferent sympathetic nerves [31], the antioxidative role of USP2 might be common to several types of cells.

In this study, NAC treatment restored mitochondrial ATP synthesis in both *Usp2*KO and ML364-treated C2C12 cells, indicating that oxidative stress contributes to impaired oxidative phosphorylation in USP2-deficient cells. In addition to defects in ATP synthesis, USP2-deficient C2C12 cells also showed negative impacts on mitochondria [33]. To date, USP2 has been shown to stabilize mitofusin 2 (MFN2) in cardiomyocytes [43]. MFN2, a GTPase on the outer mitochondrial membrane, is a prerequisite for mitochondrial fusion [44]. Mitochondria dynamics, consisting of fission and fusion, form a counterbalance mechanism that accommodates mitochondria to energy requirements [45]. Thus, the decrease in MFN2 might also affect mitochondrial function in USP2-deficient myoblasts.

Some USPs confer resistance against oxidative stress via the induction of antioxidative enzymes. For instance, USP18 induces SOD and catalase to reduce malondialdehyde in pulmonary endothelial cells [46]. Alternatively, USP30 participates in the autophagy of peroxisome (so called “pexophagy”), indicating its role in intracellular ROS levels [47]. To determine the molecules responsible for USP2-dependent antioxidation, we performed the mRNA screening of 22 antioxidative molecules. Since only UCP2 was negatively influenced by both the genetic and chemical inhibition of USP2, the molecular mechanisms underlying USP2-dependent antioxidation seem to be relatively selective. However, we only checked the expression of 22 molecules at the mRNA level. Therefore, USP2 might control these molecules at post-transcriptional or post-translational level, although USP2 did not modulate the overall activities of SOD, GR, Gpx, and TrxR. Additionally, USP2 might regulate other antioxidative or ROS-generating molecules which were not examined in this study. Previously, USP2 was preferably localized to peroxisomes in hepatocytes, suggesting that USP2 might control ROS generation through pexophagy [48]. Further investigation might reveal additional factors participating in USP2-dependent antioxidation.

Of 22 genes, USP2 preserves *Ucp2* expression in C2C12 cells. UCP2 dampens ROS generation and subsequent metabolic defects, including diabetes mellitus, atherosclerosis, and cardiomyopathy [14,49]. Thus, impaired expression of *Ucp2* may cause ROS accumulation in USP2-deficient C2C12 cells. This theory was supported by introducing a *Ucp2*-expressing construct conferring resistance against mitochondrial dysfunction in *Usp2*KO C2C12 cells. Unlike the restricted expression of UCP1 and UCP3 in brown adipose tissue and skeletal muscle, UCP2 is widely distributed across various tissues or cells, including muscle, adipose tissue, brain, pancreatic islet, and macrophages [17]. Considering that the tissue distribution of USP2 is also relatively ubiquitous [50], USP2 might prevent oxidative stress in various tissues through the preservation of UCP2.

The lack of USP2 dramatically repressed UCP2 at the mRNA level, implying that USP2 stimulates the transcription of the *Ucp2* gene. In an early study, Spiegelman and colleagues demonstrated that the overexpression of PGC1α increased the *Ucp2* mRNA levels in mature C2C12 cells, suggesting that PGC1α transcriptionally increases *Ucp2* expression [39]. Since USP2 delayed the degradation of PGC1α, USP2 might maintain UCP2 levels due to the stabilization of PGC1α. Indeed, the overexpression of the *Usp2* gene decreased the quantity of K48-linked polyubiquitin chains on PGC1α, suggesting that USP2 is a direct stabilizing enzyme for PGC1α through deubiquitination. A recent paper reported that PGC1α repressed the accumulation of mitochondrial ROS and subsequent DNA damage in myoblasts, preventing sarcopenia [51]. Therefore, USP2 might impede sarcopenia through the preservation of PGC1α.

In this study, USP2 deficiency caused a remarkable decrement in *Ucp2* mRNA in parallel with the decrease in the PGC1α protein. Since we did not demonstrate the involvement of PGC1α in USP2-modulated UCP2 expression, we cannot exclude the possibility that USP2 promotes *Ucp2* expression via other transcriptional regulators. It has been suggested that PPARs, FOXA1, and SMAD4 regulate *Ucp2* expression [52]. Additionally, the promoter region of the human *UCP2* gene has several *cis*-regulatory elements, including the specific protein-1 (Sp1) binding site, the sterol regulatory elements, and the thyroid hormone response elements, indicating that Sp1, sterol binding proteins (SRBPs), and thyroid receptor (TR) can control *UCP2* expression [18]. Notably, PGC1α potentiates the expression of SREBP-1c and SREBP2 in β-cell-like INS-1E cells, implying that PGC1α can indirectly maintain *Ucp2* expression via SREBP induction [53].

The current study indicates that the UCP2 protein levels were not increased in *Ucp2*-introduced C2C12 cells, although these cells exhibited a significant increase in *Ucp2* mRNA levels. It has been reported that UCP2 is rapidly degraded by the cytosolic ubiquitin–proteasome system in several cells [54]. Notably, the degradation of the UCP2 protein in INS-1E cells was controlled by the concentration of glutamine or glucose, although the nutrients did not affect *Ucp2* mRNA levels [55]. Additionally, the half-life of the UCP2 protein varied among the organs of rats [56]. These observations imply that post-translational control might be a crucial determinant of UCP2 protein levels. Based on this idea, a putative protein digestion mechanism might reduce the excessive accumulation of UCP2 in C2C12 cells.

As found in the current study, the inhibition of USP2 interrupts mitochondrial ATP synthesis in myoblasts, leading to defects in proliferation and differentiation [33,42]. In contrast, *Usp2*KO mice typically develop without any defects in skeletal muscle [57]. The difference between cellular and animal models might be attributed to the degree of oxidative stress. Most progenitors of myocytes in tissues remain quiescent under physiological conditions, while cultured myoblasts continuously proliferate. To obtain sufficient ATP for proliferation, hematopoietic stem cells in the fetal liver promote oxidative phosphorylation and subsequent ROS generation more than those in bone marrow [58]. Similarly, cultured myoblasts might expedite oxidative phosphorylation compared with myocyte progenitors in muscle, bringing about the accumulation of mitochondrial ROS. Furthermore, cultured cells are usually maintained at atmospheric oxygen levels, while oxygen concentrations in tissues range between 2 and 9% [59]. Since 1–4% of the oxygen consumed by the mitochondria is deflected to produce ROS, an excess of ROS are produced in cultured cells compared with cells within the body under normal physiological conditions [59]. Therefore, cultured myoblasts might be liable to express oxidative mitochondrial damage owing to USP2 deficiency. Given this, USP2-dependent mitochondrial defects in myoblasts are likely to become more apparent in the skeletal muscle of individuals suffering from oxidative stress, such as diabetes mellitus patients.

## 4. Materials and Methods

### 4.1. Cells

Mouse myoblast C2C12 cells were obtained from the RIKEN Bioresource Center (Tsukuba, Japan). *Usp2*KO C2C12 clone #1 has previously been documented [33]. Human embryonic kidney 293FT cells were purchased from Thermo Fisher Scientific (Waltham, MA, USA). All cells were cultured in 4.5 g/L glucose-containing Dulbecco’s modified Eagle medium supplemented with 10% FCS. In some experiments, C2C12 cells and their derivatives were treated with 10 μM ML364 (MedChem Express, Monmouth Junction, NJ, USA) for 8 h. For the removal of mitochondrial ROS, cells were incubated in the presence of 5 mM NAC (Fujifilm Wako, Osaka, Japan) for 8 h. Cells were treated with 100 μg/mL cycloheximide (Fujifilm Wako) to prevent nascent protein synthesis. To inhibit the proteasome, 10 μM MG132 (TCI, Tokyo, Japan) was supplemented for 5 h.

### 4.2. Plasmid Construction and Transfection

The construction of phUSP2-EGFP and phUSP2-HA plasmids was performed as previously described [29]. pEGFP-N2 (Clontech, Mountain View, CA, USA) and pcDNA3-HAC (provided by Dr. Hiroyuki Takatsu, Kyoto University, Kyoto, Japan) were used as “empty plasmid” controls. pFN21A-HaloTag-hPGC1α and pFN21A-HaloTag-CMV Flexi plasmids were purchased from Promega (Madison, WI, USA). HA-ubiquitin was obtained from Addgene (Watertown, MA, USA). A point mutation to substitute cysteine with alanine at the 276th amino acid of human USP2A (NP_004196) was performed by using a KOD-Plus-Mutagenesis kit (Toyobo, Osaka, Japan). The plasmids were transfected into 293FT cells by using Fugene HD reagent (Promega).

### 4.3. Lentivirus Constructs and Infection

The coding region of mouse *Ucp2* (accession No. BC012697) was cloned into a pDONR221 vector (Thermo Fisher Scientific) by a Gateway BP reaction and subsequently converted into pLenti6/V5-DEST (Thermo Fisher Scientific) by a Gateway LR reaction. The DNA sequence of the pLenti6-m*Ucp2*-DEST construct was checked by Hokkaido System Science (Sapporo, Japan). Recombinant lentiviral particles were produced in 293FT cells by using ViraPower Lentiviral Packaging Mix (Thermo Fisher Scientific). Lentivirus particles derived from the transfected cells were concentrated by using Lenti-X Concentrator (Takara Bio, Otsu, Japan) and used to infect C2C12 cells. Infected cells were selected by using 1–2 μg/mL Blasticidin S (InvivoGen, Hong Kong, China).

### 4.4. Mitochondrial ROS Accumulation

Mitochondrial ROS were visualized by treatment with MitoSOX Red superoxide indicator (Thermo Fisher Scientific). The cells were stained with 5 μM MitoSOX Red reagent for 20 min. After washing with PBS twice, the cells were monitored by using FACS Verse (BD Biosciences, Franklin Lakes, NJ, USA) or by a BZ-H4A microscope (Keyence, Osaka, Japan). The nuclei were stained with 5 ng/mL Hoechst33342 (Thermo Fisher Scientific). The quantitative analysis of flow cytometry was performed by using FACS Diva (BD Biosciences).

### 4.5. Mitochondrial Membrane Potential

The electric potential of the mitochondrial inner membrane was visualized by using TMRM (Setaresh Biotech, Eugene, OR, USA). The cells were incubated in the presence of 20 nM TMRM for 30 min. After washing with PBS twice, the cells were subjected to analysis by using FACS Verse flow cytometry.

### 4.6. Intracellular ATP Content

Intracellular ATP content was measured by using an ATP measurement solution (Toyo B-Net, Tokyo, Japan). After adding an equal volume of ATP measurement solution to the culture medium, the cells were vigorously mixed for 1 min and subsequently incubated at room temperature in a dark place. The chemiluminescence of the cell lysate was then measured by using a NIVO multimode microplate reader (Perkin Elmer, Waltham, MA, USA).

### 4.7. Cellular Toxicity

The cellular toxicity of ML364 was validated by monitoring LDH content in the culture supernatant. LDH content was measured by using a Cytotoxicity LDH assay kit (Dojindo, Kumamoto, Japan) according to the manufacturer’s instructions. Absorbance at 490 nm was measured by using an iMark Microplate Absorbance Reader (Bio-Rad, Hercules, CA, USA).

### 4.8. RT-qPCR Analysis

Total RNA was extracted with RNAiso Plus reagent (Takara Bio). cDNA was synthesized by using M-MLV reverse transcriptase (Nippon Gene, Tokyo, Japan). Quantitative PCR was performed by using the KAPA SYBR Fast qPCR kit (KAPA Biosystems, Wilmington, MA, USA), GeneAce SYBR qPCR Mix α, or GeneAce SYBR qPCR Mix II (Nippon Gene) by using an ECO qPCR system (Illumina, San Diego, CA, USA). mRNA levels were determined by the ∆∆Ct method, using *Hprt1* as a reference control. Dual-labeled probes were synthesized by using Sigma Genosys (Ishikari, Japan). The sequences of primers and the probe for qPCR are listed in Appendix A. Heatmaps were created by using Heatmapper (http://www.heatmapper.ca (accessed on 8 July 2024)).

### 4.9. Antioxidative Enzyme Activities

The activities of SOD, GR, Gpx, and TrxR were assessed by using commercially available kits (Cayman Chemical, Ann Arbor, MI, USA) and following the manuals’ instructions. Absorbance was measured by using a Multiskan Sky High microplate spectrophotometer (Thermo Fisher Scientific).

### 4.10. Western Blot Analysis

Western blot analysis was conducted as previously described [60]. Total cell lysates were obtained by using RIPA buffer [25 mM Tris-HCl, 150 mM NaCl, 1% Nonidet P-40, 1% sodium deoxycholate, and 0.1% SDS]. Nuclear protein was extracted by using a Nuclear Extraction kit (Active Motif, Carlsbad, CA, USA). After electrophoresis in 7.5, 10, or 15% SuperSep acrylamide gel (Fujifilm Wako), the protein was transferred onto an immobilon-P membrane (Merck Millipore, Billerica, MA, USA). After blocking with Blocking One solution (Nacalai Tesque, Kyoto, Japan), the membrane was reacted with 1000–2000-fold diluted primary antibodies against USP2 (#AP2131a; Abgent, San Diego, CA, USA), UCP2 (#CSB-PA299698; Cusabio, Houston, TX, USA), PGC1 (#ab54481; Abcam, Cambridge, UK), Halo-tag (#G921A; Promega), HA-tag (#2367 and #3724; Cell Signaling Technology, Danvers, MA, USA), K48-linked polyubiquitin (#EP8589; Abcam), GFP (#598; MBL, Tokyo, Japan), β-actin (#010-27841; Fujifilm Wako), GAPDH (#016-25523; Fujifilm Wako), RNA polymerase II (#SC-899X; Santa Cruz Biotechnology, Dallas, TX, USA), lamin-A/C (#10298-1-AP; Proteintech, Rosemont, IL, USA), and γ-tubulin (#sc-17787; Santa Cruz Biotechnology) at 4 °C overnight. Then, 5000-fold diluted horseradish-conjugated anti-rabbit (#7074; Cell Signaling Technology) or anti-mouse (#7076; Cell Signaling Technology) immunoglobulin was used as a secondary antibody. Primary and secondary antibodies were diluted with Can-Get-Signal enhancer solution (Toyobo) or Tris-buffered saline with Tween 20 (TBS-T: 50 mM Tris-HCl, 140 mM NaCl, 2.7 mM KCl, and 0.5% Tween 20; pH 7.6) supplemented with 10% Blocking One solution (for β-actin and γ-tubulin). The immune complex was visualized by using Chemilumi One Super reagent (Nacalai Tesque) and scanned by using Gene Gnome 5 (Syngene, Cambridge, UK). Quantification analysis was performed by using GeneTools software version 4.3.17.0 (Syngene).

### 4.11. Immunoprecipitation

The immunoprecipitation of Halo-tagged PGC1α was performed by using Halo-Trap Magnetic Particles M-270 (Chromotek, Planegg, Germany) according to the manual’s instructions with a minor modification. The beads were washed with a washing buffer containing 10 mM Tris-HCl (pH 7.5), 150 mM NaCl, 0.5 mM EDTA, and 0.5% Nonidet P40.

### 4.12. Statistical Analysis

Statistical analyses were performed by using a Student’s *t*-test (for two groups) or one-way analysis of variance followed by Tukey’s post hoc test (for more than three groups) by using KaleidaGraph software version 4.5J (Synergy Software, Reading, PA, USA).

## 5. Conclusions

USP2 prevents mitochondrial dysfunction by mitigating oxidative stress in culture myoblasts. Mechanistically, the protective role of USP2 may be attributed to the maintenance of UCP2 expression. Since USP2 stabilizes PGC1α by deubiquitination, USP2 might eliminate mitochondrial ROS by activating the PGC1α–UCP2 axis.

## Figures and Tables

**Figure 1 ijms-25-11936-f001:**
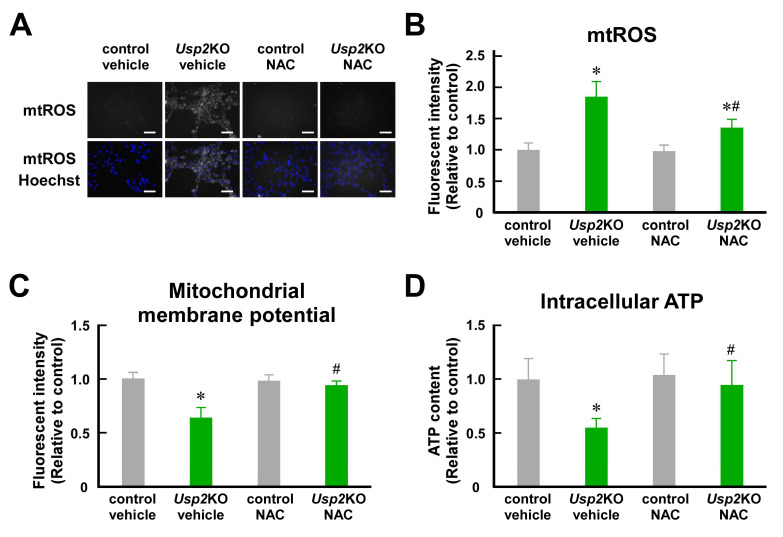
ROS are involved in *Usp2* deficiency–elicited mitochondrial dysfunction in C2C12 myoblasts. Control and *Usp2*KO C2C12 cells were treated with 5 mM NAC or a vehicle (2 mM DMSO) for 8 h. (**A**,**B**) The accumulation of mitochondrial reactive oxygen species (ROS). Mitochondrial ROS (mtROS) were visualized by using MitoSOX Red (Thermo Fisher Scientific, Waltham, MA, USA). The fluorescent signal is shown in gray. Representative microscopic images of three experiments are shown (**A**). The nuclei were stained with Hoechst33342 (Thermo Fisher Scientific). All of the scale bars represent 50 μm. The mean intensity of the MitoSOX Red-derived fluorescent signal was measured by using a flow cytometer (**B**). (**C**) The mitochondrial membrane potential was visualized by using TMRM (Setaresh Biotech, Eugene, OR, USA) staining. (**D**) Intracellular ATP content. All values are relative to the mean ± SD of the vehicle-treated control C2C12 cells (**B**–**D**). The data are the means of 6 wells (**B**–**D**). * *p* < 0.05 vs. control; ^#^
*p* < 0.05 vs. vehicle-treated group.

**Figure 2 ijms-25-11936-f002:**
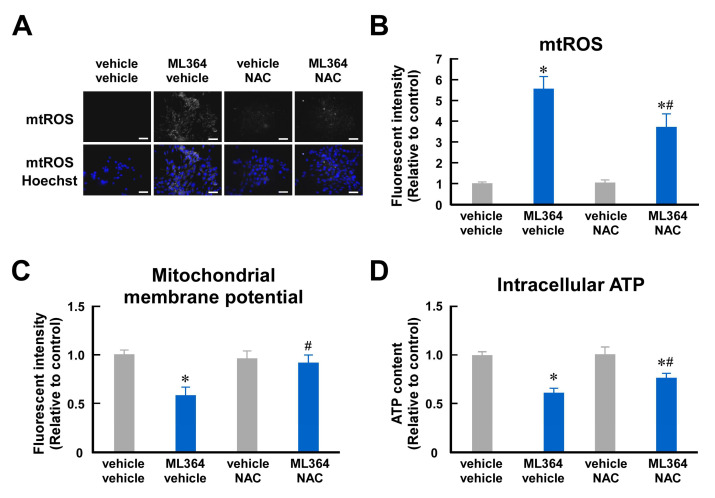
ROS are involved in ML364-elicited mitochondrial dysfunction in C2C12 myoblasts. C2C12 cells were treated with 10 μM ML364 or a vehicle (2 mM DMSO) for 8 h, and 5 mM NAC or the vehicle (2 mM DMSO) were added at the same time as ML364 or vehicle application. (**A**–**D**) Mitochondrial reactive oxygen species (mtROS) (**A**,**B**), mitochondrial membrane potential (**C**), and intracellular ATP content (**D**) were analyzed as per Figure 1. All of the scale bars represent 50 μm (**A**). All values are relative to the mean of the vehicle-treated C2C12 cells (**B**–**D**). The data are the means ± SDs of 6 wells (**B**–**D**). * *p* < 0.05 vs. ML364-untreated group; ^#^
*p* < 0.05 vs. NAC-untreated group.

**Figure 3 ijms-25-11936-f003:**
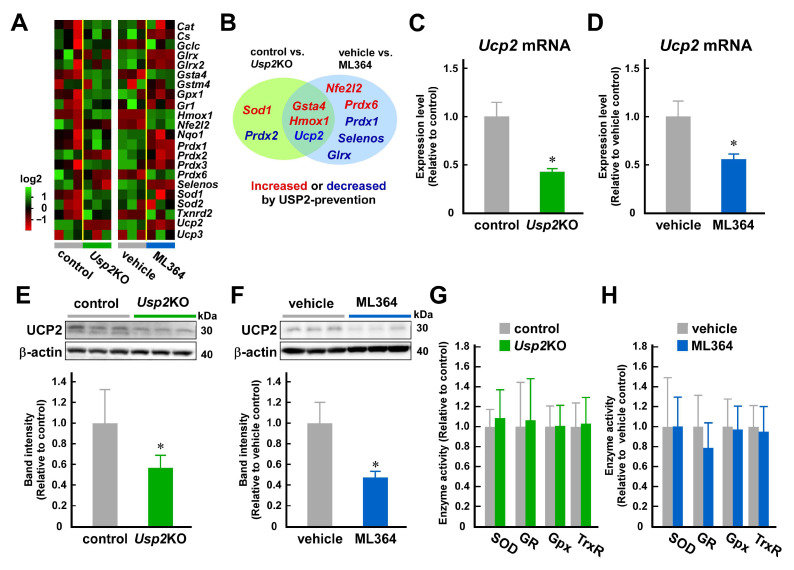
USP2 fosters UCP2 expression in C2C12 myoblasts. (**A**) The RT-qPCR screening of canonical antioxidative genes. Two comparisons were carried out: control and *Usp2*KO C2C12 cells (**left**) and treatment with vehicle and ML364 (10 μM, 8 h) (**right**). Heat maps represent the log2 expression ratio to the means of the control (**left**) or vehicle-treated cells (**right**). (**B**) A Venn diagram of genes whose expression was affected by the genetic or chemical inhibition of USP2. (**C**,**D**) The abundance of *Ucp2* mRNA in control and *Usp2*KO C2C12 cells (**C**) and vehicle- or ML364-treated C2C12 cells (**D**). (**E**,**F**) UCP2 protein levels. Western blot analysis was performed by using total cellular lysate of control and *Usp2*KO C2C12 cells (**E**) and vehicle- or ML364-treated C2C12 cells (**F**). Representative images are shown (**top**). (**G**,**H**) The activities of SOD, GR, Gpx, and TrxR in *Usp2*KO C2C12 cells (**G**) and ML364-treated C2C12 cells (**H**). Expression values were normalized to Hprt1 mRNA levels (**A**,**C**,**D**) or β-actin levels (**E**,**F**). All values are relative to the means of control C2C12 cells (**C**,**E**,**G**) or vehicle-treated C2C12 cells (**D**,**F**,**H**). The data are the means ± SDs of 6 (**C**–**E**,**G**,**H**) or 3 (**F**) wells. * *p* < 0.05 vs. control (**C**,**E**) or vehicle-treated groups (**D**,**F**).

**Figure 4 ijms-25-11936-f004:**
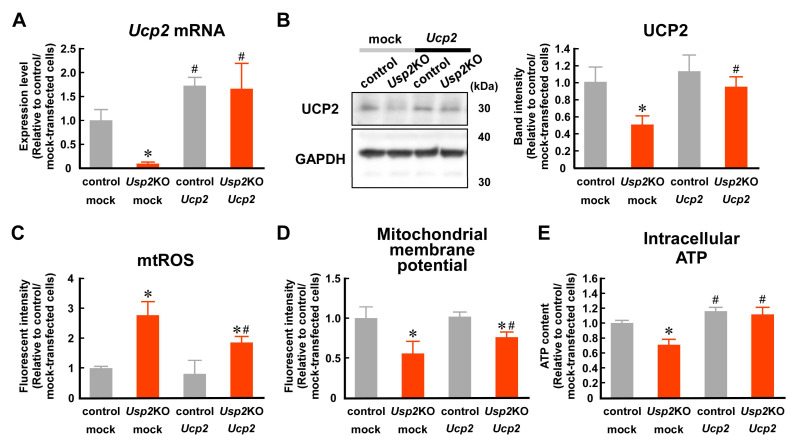
The introduction of UCP2 partially recovers *Usp2*KO-elicited mitochondrial dysfunction. Control or *Usp2*KO C2C12 cells were infected with *Ucp2*-expressing or mock lentiviral particles. (**A**,**B**) mRNA (**A**) and protein (**B**) levels of UCP2. A representative image is shown (**B**, left). The expression values were normalized to *Hprt1* mRNA levels (**A**) or GAPDH levels (**B**). (**C**,**D**) Mitochondrial ROS and mitochondrial membrane potential were measured by flow cytometry. (**E**) Intracellular ATP content. All values are relative to the means of mock-transfected control C2C12 cells. The data are the means ± SDs of 4 (**A**), 3 (**B**), or 6 (**C**–**E**) wells. * *p* < 0.05 vs. control C2C12 cells; ^#^
*p* < 0.05 vs. mock-transfected group.

**Figure 5 ijms-25-11936-f005:**
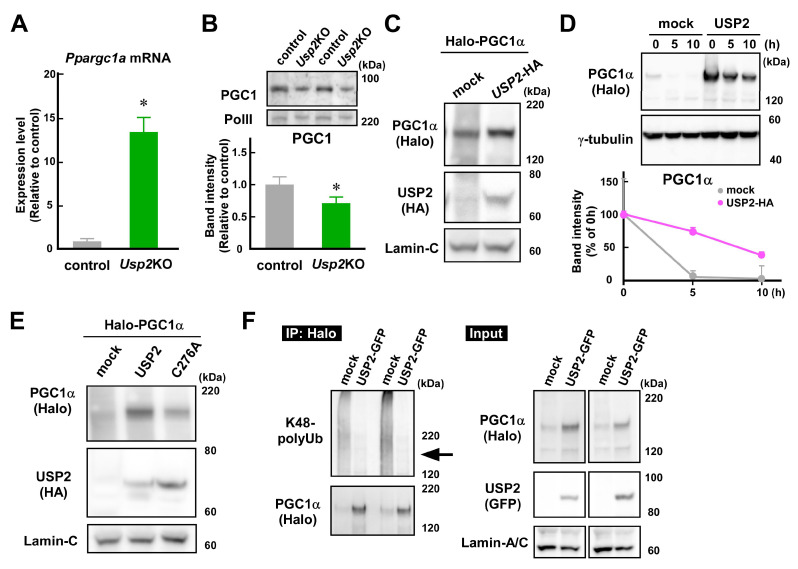
USP2 prevents the degradation of PGC1α. (**A**) The expression levels of the *Ppargc1a* gene in control and *Usp2*KO C2C12 cells. *Ppargc1a* expression was normalized to that of *Hprt1*. (**B**–**F**) Nuclear proteins were extracted from *Usp2*KO and control C2C12 cells (**B**) and 293FT cells transfected with cDNA-expressing constructs (**C**–**F**). (**B**) The nuclear contents of PGC1 in control and *Usp2*KO C2C12 cells. PGC1 levels were normalized to RNA polymerase II (PolII) levels (bottom). (**C**) The co-transfection analysis of USP2 and PGC1α. Halo-tagged PGC1α and HA-tagged USP2 or empty plasmids were transfected into 293FT cells. The nuclear fraction was subjected to Western blot analysis two days after transfection. (**D**) The effects of USP2 on the stability of PGC1α. Two days after transfection with Halo-tagged PGC1α, HA-tagged ubiquitin (Ub), and GFP-tagged USP2, or GFP (mock) plasmids in 293FT cells, cycloheximide was added to reach 100 μg/mL. The nuclear fraction was extracted at the indicated times. The data are represented as % intensities of cycloheximide-untreated cells. (**E**) Roles of isopeptidase activity of USP2 in PGC1α stability. Halo-tagged PGC1α and HA-tagged USP2, isopeptidase-mutated USP2 (C276A), or empty plasmids (mock) were transfected for one day. Cells were treated with 100 μg/mL cycloheximide for 5 h. (**F**) The effects of USP2 on the K48-linked polyubiquitination of PGC1α. The nuclear extract of 293FT cells, which were transfected with Halo-tagged PGC1α and HA-tagged ubiquitin plasmids in combination with GFP-tagged USP2 or GFP (mock) plasmids, was treated with 10 μM MG132 for 5 h and then subjected to immunoprecipitation against Halo-tag. Western blot images of input samples are also shown. An arrow indicates the size of Halo-tagged PGC1α (**F**). For the loading control, PolII (**B**), lamin-A/C or C (**C**,**E**,**F**), or γ-tubulin (**D**) was detected. A representative image of 3–4 replicates is shown (**B**–**F**). The data are the means ± SDs of 4 wells (**A**,**B**,**D**). The data were obtained from two membranes (**D**). Differences in band intensities between the membranes were normalized to the sum of the band intensities of each replicate (**D**). * *p* < 0.05 vs. mock-transfected control (**A**,**B**).

## Data Availability

All data generated or analyzed during this study are included in this manuscript and additional supporting files, which can be downloaded at https://www.dropbox.com/scl/fo/qc2ppx48z7dzv91mux5mx/AEz3AbcTpd_jEE3RV_8nxe8?rlkey=vf5fj5bdmm4mx0kv63rk7urxz&st=642qvhaa&dl=0 (accessed date 29 July 2024).

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
