# Peer review of "USP2 Mitigates Reactive Oxygen Species-Induced Mitochondrial Damage via UCP2 Expression in Myoblasts"

_ijms, 2024, doi:10.3390/ijms252211936_

Round 1

Reviewer 1 Report

Comments and Suggestions for Authors

In this paper Kitamura et al demonstrated the importance of mitochondrial integrity by Usp2 gene, whose regulation is mediated by ucp2-pgc1alfa.

I have  major points, some of them very important, for whose I think that the manuscript cannot be accepted in the present form.

1-why authors showed the effect of Usp2 KO in myoblasts and not in myotubes? Have authors tried to differentiate c2c12 cells and assess the effect of USP2 depletion? This is very important, because such data probably conducted on myoblast regards the physiology only of progenitor cells (satellite cells) and not of myotubes. It should be important to demonstrate the effect of Usp2 KO in myotube and their mitochondrial functionality.

2- In the RT-PCR experiments  (fig 3a), if i understood correctly data were not normalized to a control transcript (such as gapdh, histone). I suggest taking in consideration such observation, the great variability observed in the control could be due to the lack of such normalization

3-Change the blot reported in figure (fig 3e), it is clear that the amount of b.actin is lower in uspko cells. I saw the raw blot and other lanes are better than that reported in the figure.

4- I think that the overexpression of UCP2 didn't work so well in figure 4, (overexpressing cells are similar to the control) probably it is a protein difficul to overexpress. But I have a question. Why you need to infect c2c12 cells? usually such cells are transfected very well with lipofectamine. I suggest to repeat the experiment with a method that allow a higher expression of the protein. Also, in the raw blot of that figure there are strong bands at approximately 40 kDa? is it GAPDH? if it is the case, the second development with GAPDH is clearly oversaturated and authors have to consider again the quantification. 

6-I have big doubts about figure 5b raw blot, it is clear that the two blots shown are exactly the same with a different exposure but the mw are shift down in the second one (tubulin). If what I think is correct, the band that author are showing as tubulin, is a protein that in the first blot is of 30 kda while in the second blot of 50. 

Reviewer 2 Report

Comments and Suggestions for Authors

In their manuscript, Kitamura and colleagues investigate the role of the deubiquitinating enzyme USP2 in protecting myoblasts against ROS-induced mitochondrial damage. They first show that USP2 dysfunction is associated with mitochondrial ROS and mitochondrial dysfunction. They then investigate the connection between USP2, UCP2, and PGC1alpha in this process. Overall, the study is interesting and brings new mechanistic elements. However, some questions may be considered by the authors.

Major comments:

1. It seems that the catalytic activity of USP2 is necessary because the inhibitor ML364 produces effects similar to those observed in Usp2KO cells. However, conducting a rescue experiment in the Usp2KO cell line would provide stronger evidence for this conclusion.

2. In Figures 5B, 5C, and 5D, the authors refer to "nuclear content" in the western blot and quantification. It's important to clarify whether the samples are from nuclear extraction or whole cell lysates. Additionally, the authors should include the cytosolic or whole cell lysate content alongside the nuclear content with appropriate loading control, such as lamin for the nuclear fraction, as tubulin is a control for whole cell lysate/cytosolic content, not nuclear content.

3. In Figure 5F, the molecular weights are missing on the Ub blots. It is expected that the smear of ubiquitinated PGC1 should be above Halo-PGC1. However, it seems that the smears on the IP lanes extend as in the input lanes, indicating that the authors detect ubiquitination of a broad range of proteins, rather than being specific for PGC1. Since USP2 is known for having numerous partners and substrates, the authors should consider performing the PGC1 immunoprecipitation under denaturing conditions to eliminate other partners of PGC1 that may also be affected by USP2.

4. Finally, does the overexpression of PGC1 in Usp2KO cells rescue the ROS and mitochondrial defects through Ucp2 expression?

Minor comments:

1. Fig1A, -2A: Using grey levels instead of red could improve visualization. 

2. In figure legends, authors should specify if the error bars are SD or SEM.

Round 2

Reviewer 2 Report

Comments and Suggestions for Authors

In the revised version of their manuscript and rebuttal letter, the authors have answered all my points. A few new experiments have been conducted. Where technical difficulties arose (immunoprecipitation of PGC1a under denaturing condition), the authors acknowledged the issues they encountered in the rebuttal letter and tuned down their conclusion in the discussion of their results.

Minor points:

- The western blots are still missing the molecular weight information across the manuscript. They are in the original pictures of the membranes, however MW should also be displayed in the main figures.

- Original images for Figure 5B display PGC1 and g-Tubulin; however in Figure 5B the labels indicate PGC1 and PolII. The authors should correct.

- Figure 5C: the shape of the bands or their intensity doesn't seem to match that of the "original images". If the wrong images were used for crop or original, this should be corrected.

- Figure 5F: same comment as for 5C.

- 4.11. "immune precipitation" -> "immunoprecipitation"?

Comments on the Quality of English Language

Some typos may remain. For instance:

Line 255: "nuclear protein was extracted" -> nuclear proteins were extracted"

Line 278: do the authors mean "It was previously shown that L6..."?
